# Incidence, Risk Factors and Clinical Implications of Glucose Metabolic Changes after Heart Transplant

**DOI:** 10.3390/biomedicines10112704

**Published:** 2022-10-26

**Authors:** Emanuele Durante-Mangoni, Domenico Iossa, Valeria Iorio, Irene Mattucci, Umberto Malgeri, Daniela Pinto, Roberto Andini, Ciro Maiello, Rosa Zampino

**Affiliations:** 1Unit of Infectious and Transplant Medicine, AORN Ospedali dei Colli-Monaldi Hospital, Piazzale Ettore Ruggieri Snc, 80131 Naples, Italy; 2Department of Precision Medicine, University of Campania ‘Luigi Vanvitelli’, 80138 Naples, Italy; 3Unit of Heart Transplant, AORN Ospedali dei Colli-Monaldi Hospital, 80131 Naples, Italy; 4Department of Advanced Medical and Surgical Sciences, University of Campania ‘Luigi Vanvitelli’, 80138 Naples, Italy

**Keywords:** diabetes mellitus, transplant, complications, mortality, outcome, glucose metabolism, body mass index, metabolic syndrome

## Abstract

Diabetes mellitus (DM) arising *de novo* after transplant is a common complication, sharing many features with type 2 DM but also specific causes, such as administration of steroids and immunosuppressive drugs. Although post-transplant DM (PTDM) is generally assumed to worsen recipients’ outcomes, its impact on renal function, cardiac allograft vasculopathy and mortality remains understudied in heart transplant (HT). We evaluated incidence and risk factors of PTDM and studied glucose metabolic alterations in relation to major HT outcomes. 119 subjects were included in this retrospective, single centre, observational study. A comprehensive assessment of glucose metabolic state was done pre-transplant and a median of 60 months [IQR 30–72] after transplant. Most patients were males (75.6%), with prior non-ischemic cardiomyopathy (64.7%) and median age of 58 years [IQR 48–63]. 14 patients developed PTDM, an incidence of 3.2 cases/100 patient-years. Patients with worsening glucose metabolic pattern were the only who showed a significant increase of BMI and metabolic syndrome prevalence after transplant. 23 (19.3%) patients died during follow up. Early mortality was lower in those with stably normal glucose metabolism, whereas improvement of glucose metabolic state favorably affected mid-term mortality (log-rank *p* = 0.028). No differences were observed regarding risk of infections and cancer. PTDM is common, but glucose metabolism may also improve after HT. PTDM is strictly related with BMI increase and metabolic syndrome development and may impact recipient survival.

## 1. Introduction

Metabolic disorders, particularly diabetes mellitus (DM), are among the most common and serious complications observed in heart transplant (HT) recipients [1,2,3]. Often a pre-transplant comorbidity, DM arising *de novo* after transplant, previously referred to as *new onset diabetes after transplantation* [4], is now defined as *post-transplant diabetes mellitus* (PTDM). Specific criteria are in place to identify and classify the emergence of PTDM [2,5]. Absence of DM before transplant and HbA1c values defining DM have been added to the ADA criteria [6,7] for the diagnosis of PTDM.

PTDM prevalence ranged between 17% and 30% in recent studies [2,8,9,10]. PTDM shares with type 2 DM many common causes (age, body mass index [BMI], family history, metabolic syndrome, ethnicity) [2,11,12] but also presents specific, transplant-related causes, such as administration of steroids and different immunosuppressive drugs, including calcineurin inhibitors and m-TOR inhibitors [13,14,15,16,17].

Although PTDM can worsen the outcome of recipients over time, increasing their cardiovascular risk [1], available data did not show a clear association with Cardiac Allograft Vasculopathy (CAV) or other signs of graft dysfunction [18,19]. Also, most of the knowledge on PTDM comes from kidney transplant [20] and might not be entirely applicable to different organ transplant settings. Indeed, the incidence of PTDM appears to be highest in patients receiving lung and liver transplants, intermediate in recipients of heart transplants and lowest in recipients of kidney transplants, the latter being the most studied subgroup [20]. However, PTDM is associated to higher mortality regardless of the organ being transplanted (kidney, liver, lung or heart) [21,22,23,24]. In contrast with what was observed in other transplant recipients, no specific cardiac disease has been considered as a risk factor for the development of PTDM [25]. The evaluation of glucose metabolic changes not fulfilling a formal PTDM diagnosis could also provide interesting clinical insights on cardiovascular risk and metabolic derangement after HT.

Accordingly, aims of the present study were: (i) to describe changes in glucose metabolism after HT in a large cohort of HT recipients; (ii) to evaluate the incidence and risk factors of pre-diabetes and/or PTDM in the study group; (iii) to relate glucose metabolic alterations to HT outcomes.

## 2. Patients and Methods

### 2.1. Study Design

This is a retrospective observational study aimed to detect incidence of and risk factors for PTDM in a single center cohort of HT recipients. PTDM diagnosis was made according to the current International Guidelines [ADA, Chowdry] where PTDM diagnosis is based on the same criteria of DM diagnosis in the general population.

Patients who had received HT between January 2008 and December 2017 at the Monaldi Hospital, Naples, were included in the study. Recipients of emergency transplants, for whom a specific pre-transplant evaluation was not available, were excluded.

Retrospective data collection from heart transplant recipients’ electronic records was approved by the University of Campania and AORN OspedalideiColli Ethics Committee (protocol n. 28/2019). Patient informed consent was waived.

### 2.2. Pre-Transplant Assessment

A comprehensive pre-transplant screening was performed, according to a predefined clinical protocol in use in our centre. It included collection of clinical history data, a general physical examination, and hematochemical and imaging studies.

For each patient, we collected and stored in an electronic database age, sex, current drug therapy, arterial blood pressure, heart rate, body weight, height and waist circumference. At history taking, we assessed the etiology of dilative cardiomyopathy, the presence of a positive family history of DM, the presence of atherosclerotic cardiovascular disorders/events, arterial hypertension and DM.

Systolic blood pressure was classified according to the ESC guidelines [26]. BMI, computed as the ratio between the body weight (in kilograms) and the square of the height (in meters), allowed patients’ classification as underweight (<18.5 kg/m^2^), normal (M: 18.5–24.9 kg/m^2^; F: 18.5–23.9 kg/m^2^), overweight (M: 25–29.9 kg/m^2^; F: 24–29.9 kg/m^2^), and obese: 1st degree (30–34.9 kg/m^2^), 2nd degree (35–39.9 kg/m^2^), 3rd degree obesity (≥40 kg/m^2^). Waist circumference (WC) was measured and used as an indicator of central/visceral adiposity. In particular, as all patients studied were of European descent, a WC >88 cm and >102 cm among women and men, respectively, were considered as visceral obesity diagnostic cut-offs. Furthermore, presence of Metabolic Syndrome (MetS) was defined for each patient, according to ATP III criteria [27].

Among hematochemical parameters, all measured by our hospital central laboratory with standard techniques, we considered: fasting glycemia, glycated hemoglobin (HbA1c), LDL, HDL and total cholesterol, and triglycerides. Creatinine was measured and used to assess the estimated glomerular filtration rate (eGFR) according to the CDK-EPI formula. Fasting glycemia values allowed an initial stratification of each subject in terms of possible presence of DM, impaired fasting glycemia, or normal glycemia. For fasting glycemia levels < 126 mg/dL and HbA1c <48 mmol/mol (<6.5%) with a negative DM clinical history and no anti-diabetic therapy, a standard oral glucose tolerance test (OGTT) was performed. A 2 h post-OGTT serum glucose range of 140–200 mg/dL identified impaired glucose tolerance, whilst glucose levels > 200 mg/dL were diagnostic of DM. Normal glucose tolerance was defined as a post-load glycemia value < 140 mg/dL.

### 2.3. Post-Transplant Evaluation

Patients who died during the same hospital admission when the transplantation procedure was performed were excluded from this study. Therefore, the study group involved only subjects who survived the initial hospitalization and returned as outpatients with a follow-up of at least 6 months. The maintenance immune suppressive protocol comprised a calcineurin inhibitor (either cyclosporine A or tacrolimus), an antiproliferative agent (either mycophenolate or everolimus) and prednisone, at a starting dose of 10 mg every 12 h, progressively tapered off by 2.5 mg every 4 weeks down to a flat dose of 5 mg/day life-long in the absence of specific complications. The dosage of immunosuppressive drugs was generally adapted on the bases of Therapeutic Drug Monitoring (TDM) and assessment of white blood cell counts.

Glycemic evaluation was performed every 1 to 3 months after transplant during routine outpatient assessments and irrespective of each subject clinical history. Prospective clinical and laboratory evaluations, aimed to completely reassess glucose metabolism, body composition and metabolic syndrome components, were repeated once in each transplant recipient after a median of 60 months after transplant. Measured parameters were: BMI, WC, hematochemical parameters (i.e., fasting glycemia, total cholesterol, LDL-cholesterol, HDL-cholesterol, triglycerides, and glycatedhaemoglobin). Furthermore, non-diabetic patients (fasting glycemia ≤ 126 mg/dL and HbA1c values ≤ 48 mmol/mol) underwent an OGTT to unmask possible, undiagnosed DM.

After the post-tx OGTT and by considering fasting glycemia values, patients were classified as normal, with impaired fasting glycemia, glucose intolerant or diabetics.

### 2.4. Study Outcomes

In order to study the effects of glucose metabolism alteration on HT outcomes we collected data about graft rejection (type, number of episodes), including development of graft coronary artery disease, i.e., chronic rejection. Being DM a major risk factor for kidney disease, we assessed the effect of PTDM on recipients’ renal function. In light of the known effect of DM on immune function and cell proliferation, we assessed other important post-transplant outcomes, such as occurrence of infections (>6 months post-transplant) or CMV reactivation and development of malignancies. We finally assessed the relationship between PTDM and post-transplant mortality.

### 2.5. Statistical Analysis

Statistical analyses were performed by using SPSS software 23.0. A *p*-value ≤ 0.05 was considered to denote statistical significance of the differences. Numerical data are shown as median and interquartile range (IQR) or as mean ± standard deviation (DS), whilst categorical data as number and percent. The Fisher’s exact test was used to evaluate differences in nominal variables between 2 or more groups; the Mann-Whitney U test to compare numerical variables between 2 distinct groups and the Kruskal Wallis test to evaluate more than two groups. Kaplan-Meir curves were constructed to compare survival between subgroups, and differences were assessed with the log-rank test. Logistic regression was used to analyse independent predictors of mortality.

## 3. Results

### 3.1. Clinical Characteristics of the Study Cohort

Table 1 shows the general characteristics of the 119 patients included in the study. Most were males (75.6%), with a median age equal to 58 years [IQR 48–63] at the post-transplant assessment. Evaluations were performed after a median interval of time after HT equal to 60 months [IQR 30–72], and always in the context of a phase of clinical stability in the outpatient setting. A family history of DM and cardiovascular disease was overall common.

On past clinical history, non-ischemic dilative cardiomyopathy (64.7%) was the prevalent indication for transplant, with 35.3% of patients showing a prior ischemic heart disease.

Twenty-three (19.3%) patients died during the post-transplant follow up, with a median survival time equal to 24 months [IQR 9–72] in this subgroup.

Immunosuppression maintenance hinged on cyclosporine A in more than 75% of patients, alone (3 patients) or combined with mycophenolate mophetil (MMF) or everolimus. The other patients were treated with tacrolimus combined with MMF (Table 1). Cardiovascular drugs taken during the post-transplant follow-up are also presented in Table 1.

Table 2 summarizes the most important post-transplant outcomes. There was a high rate of infectious and neoplastic complications, and most patients were hypertensive after transplant.

### 3.2. Anthropometric and Metabolic Changes after Heart Transplant

A comparison between metabolic parameters before and after HT is shown in Table 3. Despite of worsening anthropometric parameters, glucose metabolism overall improved. Specifically, after transplant, a significant reduction of IFG/IGF prevalence was observed and the number of patients showing a normal glucose profile increased (Table 3). In contrast, the prevalence of lipid metabolic changes significantly increased. MetS prevalence also rose, although not significantly (42.4% pre-transplant vs. 57.6% post-transplant; *p* = 0.062).

Comparing glucose metabolic status case-by-case before and after HT, we detected 14 patients who developed *de novo* PTDM. Indeed, based on our observations, patients with a pre-transplant normal glucose metabolism showed an incidence of PTDM equal to 3.2 cases/100 patient-years.

### 3.3. Relation of Glucose Metabolic Changes with Anthropometric and Metabolic Dysfunction

The comparison of pre- and post-transplant changes in glucose metabolism allowed us to divide patients into four subgroups. Group A: worsened, including PTDM development (N = 23); group B: improved (N = 31); group C: persistently normal glucose metabolism (N = 36); and group D: unchanged pre-diabetes/DM (N = 29). Table 4 shows the variations of metabolic parameters in these four groups. PTDM was treated with diet in all patients. In addition, oral antidiabetics (metformin, gliclazide or repaglinide) were prescribed to 21 patients, whereas insulin was administered to 15 patients, as clinically indicated.

No significant differences between the 4 groups were observed in terms of age (*p* = 0.068), heart disease etiology (*p* = 0.172) and median duration of follow-up (*p* = 0.793) (data not shown). Also, immune suppressive treatment as well as use of prednisone did not differ. Immune suppressive drug regimens in use in the 4 subgroups of patients are detailed in Appendix A. As shown, there were no significant differences across subgroups.

HDL cholesterol and triglyceride levels increased significantly in all patients. Patients with worsening glucose metabolic pattern were the only who showed a significant increase of BMI and of MetS prevalence after HT (Figure 1). LDL cholesterol did not show significant changes. Median BMI was also significantly higher in patients with worsening or stably abnormal glucose metabolism (groups A and D) compared with counterparts (groups B and C) at both pre- and post-transplant time points (data not shown).

Changes in glucose metabolic state also appeared to associate with the post-transplant dynamics of renal function. Indeed, as graphically shown in Figure 2, the eGFR showed a trend for a decrease in group A (worsening glucose metabolism) and group D (persistently altered glucose metabolism), while eGFR remained stable in groups B (improving) and C (stably normal).

Mortality was significantly different according to post-transplant glucose metabolic changes. In particular, early mortality was lower in group C patients (stably normal), whereas improvement of glucose metabolic state after transplant (group B) favorably affected mid-term mortality, which was indeed better than in group D where glucose metabolism remained altered (log-rank *p* = 0.028; Figure 3). Survival in the overall study sample (all subgroups) was also analysed in a logistic regression model in relation to all major clinical risk factors before and after transplant, as well as all variables related to glucose metabolism (Appendix A). As shown, none of the evaluated factors was an independent predictor of mortality in the multivariable analysis.

No differences were observed across the 4 groups regarding risk of infections and cancer development (not shown).

## 4. Discussion

Since the update of criteria for PTDM diagnosis, little evidence was generated regarding PTDM after HT [5]. In our study, observed incidence was substantially lower than previously described [2,28,29,30]. Stringent exclusion of pre-transplant DM and of transient post-transplant hyperglycemia likely explain the difference. Also, our evaluations were done in stable patients while on maintenance immune suppression.

HT had a clear impact on recipients’ metabolic function, associating with BMI increase, dyslipidemiam and raised MetS. In line with these data, overall DM prevalence slightly increased too. Interestingly, however, an association emerged between MetS and worsening glucose metabolism. Among patients who showed PTDM onset and any worsening of glycemic condition, a significant increase of BMI and MetS prevalence occurred, which was not observed in the other groups. This suggests that PTDM, or more broadly, glucose metabolic changes after HT are more likely linked to body composition changes rather than the effects of immune suppressive drugs. Indeed, in group B, showing improvement in glucose profiles, an inverse trend of reduction in MetS was observed.

Concurrent administration of multiple immune suppressors makes it difficult to analyse their possible role as inducers of glucose metabolic derangement [13,14,15,16,17]. PTDM pathophysiological mechanisms include factors that are specific to the post-transplant setting. Corticosteroids can promote PTDM through various mechanisms, inducing or worsening insulin resistance, increasing hepatic gluconeogenesis, reducing insulin secretion and, in the longer term, causing weight gain and visceral fat redistribution [2]. Calcineurin inhibitors reduce insulin secretion by acting on pancreatic beta cells, with tacrolimus showing a greater effect than cyclosporine [2,13]. Inhibitors of m-TOR, including everolimus, may also be implicated in the pathogenesis of PTDM, as they exert negative effects on both pancreatic beta cell function and insulin sensitivity [14,15,16]. In contrast, mycophenolic acid does not appear to affect glucose metabolism and the development of PTDM. Based on our data, we hypothesise immune suppressive agents increase overall recipient risk to develop glucose metabolic changes, with the latter manifesting phenotypically only among those characterised by an additional risk factor, e.g., when weight gain or development/worsening of MetS occurs. Immune suppressive treatment might therefore play a secondary/complementary role, as among our study subjects, these were homogeneously prescribed. Also, patient subgroups were consistent in terms of follow-up duration and, in turn, overall exposure to immune suppressive treatment.

Another interesting yet unexpected finding of our study was the positive effect HT had on glucose metabolism in a substantial subgroup of study subjects (N = 31, 26%—group B). Here, no weight gain and a trend for MetS prevalence reduction were observed. It is likely that these cases, previously affected by decompensated heart failure, recover metabolically as a consequence of improved functional capacity and more intense physical activity. Consistent with this hypothesis is the observed increase of HDL-cholesterol levels, observed in all groups, but less pronounced in those with worsening glucose metabolism. Unfortunately, we did not collect data regarding patient physical activity.

Our data suggest an impact of glucose metabolic changes on HT recipient outcomes. Diabetics who remain in this condition (group D) have a lower overall survival, whilst those showing persistently normal glucose status have the best outcome. Furthermore, an improved glucose metabolic state translates into a better mid-term survival. In contrast, and at variance with prior studies [18], there was no apparent association between PTDM or worsening glycemia and an increased occurrence of infections or malignancies.

Chronic kidney disease is a major issue in HT recipients. In our study, there were some clues to the association between this complication and a worse glucose metabolic state. Indeed, a trend for a greater reduction in the glomerular filtration rate was evident in group A, as well as in group D patients. In contrast, a role for PTDM and glucose metabolic changes in the development of HT rejection and specifically cardiac allograft vasculopathy was not evident from this study, in line with prior data [19].

In conclusion, mechanisms underlying glucose metabolic changes after HT deserve further study. Our data suggest that avoidance of weight gain and MetS development could represent a viable strategy to prevent PTDM onset. This is likely pursued by low calorie diet and physical exercise throughout the post-transplant follow-up.

## Figures and Tables

**Figure 1 biomedicines-10-02704-f001:**
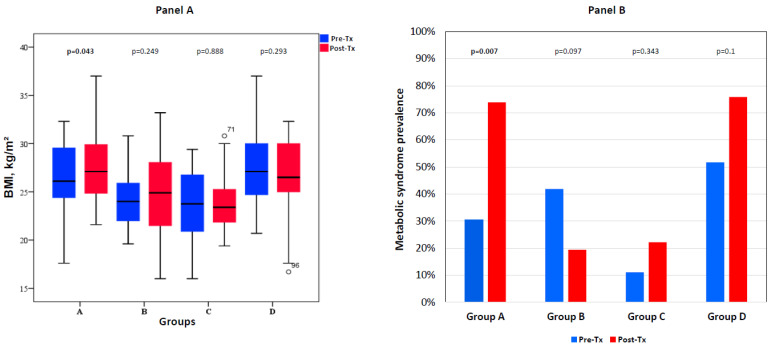
Changes in body mass index (BMI) [Panel A] and in metabolic syndrome prevalence [Panel B] from pre-transplant to post-transplant assessment in patients grouped according to changes in glucose metabolic state. Group A: worsened, including PTDM (N = 23); group B: improved (N = 31); group C: persistently normal glucose metabolism (N = 36); group D: unchanged pre-diabetes/DM (N = 29).

**Figure 2 biomedicines-10-02704-f002:**
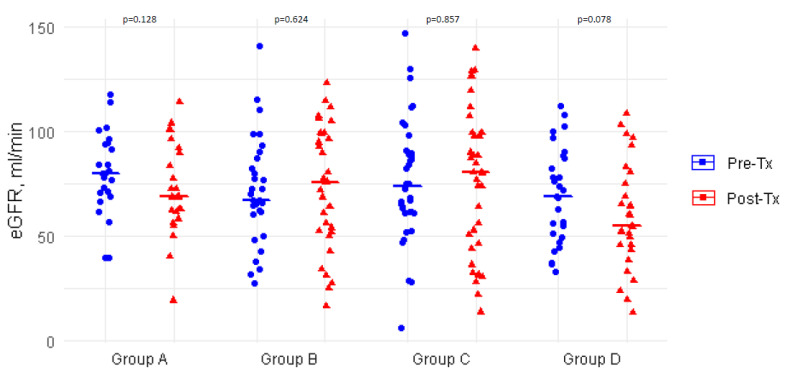
Column scatter plot showing changes in the estimated glomerular filtration rate from pre-transplant to post-transplant assessment in patients grouped according to changes in glucose metabolic state. Group A: worsened, including PTDM (N = 23); group B: improved (N = 31); group C: persistently normal glucose metabolism (N = 36); group D: unchanged pre-diabetes/DM (N = 29).

**Figure 3 biomedicines-10-02704-f003:**
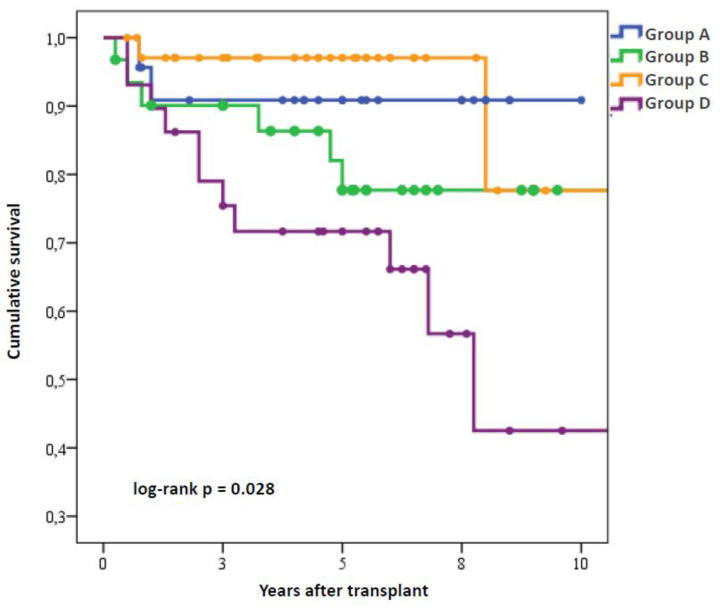
Kaplan-Meier survival curves of patients belonging to the 4 study subgroups: Group A: worsened, including PTDM (N = 23); group B: improved (N = 31); group C: persistently normal glucose metabolism (N = 36); group D: unchanged pre-diabetes/DM (N = 29).

**Table 1 biomedicines-10-02704-t001:** Clinical features of the study cohort (N = 119).

Parameter	
Age, yrs	58 [50–65]
Sex *M* *F*	90 (75.6) 29 (24.4)
Donor age, yrs	36 [23–48]
Family history of DM	38 (31.9)
Family history of CV disease	69 (58)
Family history of arterial hypertension	26 (21.8)
Smoking	61 (51.3)
History of DM pre-transplant	17 (14.3)
History of arterial hypertension pre-transplant	35 (29.4)
Years from transplant	5 [2.5–6]
Creatinine, mg/dL *pre-tx* *post-tx*	1.1 [0.9–1.4] 1.1 [0.9–1.5]
eGFR CDK-EPI, mL/min *pre-tx* *post-tx*	72.6 [57–90.3] 68.6 [50.2–96.4]
Immune suppressing agents *Cyclosporine A* *Tacrolimus* *Everolimus* *Mycophenolate*	88 (73.9) 30 (25.2) 43 (36.1) 73 (61.3)
Prednisone	99 (83.2)
Diuretics *Furosemide* *Mineralocorticoid receptor antagonists*	61 (51.3) 38 (31.9)
Beta-blockers	88 (73.9)
Anti-platelet agents	100 (84)
Oral anticoagulants	12 (10.2)
Mortality	23 (19.3)
**Pre-transplant cardiac parameters**	
Ischemic heart disease pre-transplant	42 (35.3)
Serum NT-pro BNP, pg/mL	3198 [1453–5271]
Left ventricle ejection fraction, %	25 [20–30]
Left ventricle diameter in diastole, cm	6.7 [5.9–7.5]
Cardiac Index, L/min/m^2^	2.1 [1.8–2.7]
Cardiac Output, L/min	3.9 [3.2–4.86]
Wedge pressure, mmHg	26 [18.75–31]
NYHA class *I* *II* *III* *IV*	0 (-) 20 (16.8) 78 (65.5) 21 (17.6)

Data are expressed as number (percent) or median [interquartile range-IQR].

**Table 2 biomedicines-10-02704-t002:** Post-transplant cardiovascular and non-cardiovascular outcomes.

Parameter	
Cardiac Allograft Vasculopathy	8 (6.8)
Number di cellular rejection episodes *0* *1* *2*	95 (80.5) 19 (16.1) 4 (3.4)
Neoplasia	20 (16.8)
Infections	64 (53.8)
Post-tx arterial hypertension	61 (51.3)
CMV infection *No infection/low level replication* *Clinically significant reactivation*	81 (68.1) 37 (31.1)

Data are expressed as number (percent).

**Table 3 biomedicines-10-02704-t003:** Metabolic parameters before and after transplant.

Variables	Pre-Transplant		Post-Transplant		
	N, (%)	Median (IQR)	N, (%)	Median (IQR)	*p*
Body mass index, kg/m^2^		24.9 [22–28]		25.2 [22.8–28.4]	0.022
Waist circumference, cm		98 [89–102]		102 [94–110]	0.001
Fasting glycemia, mg/dL		101 [88–122.5]	94	94 [88–108]	0.010
Glucose metabolic profile *Normal* *IFG/IGT* *Diabetes mellitus*	47 (39.5) 41 (34.5) 31 (26.1)		66 (55.5) 16 (13.4) 37 (31.1)		0.000
Cholesterol, mg/dL *Total* *HDL* *LDL*		153 [126–186] 43 [34.75–51] 93 [70.75–118]		184 [148–217] 54.5 [44–68] 100.5 [74.7–123.8]	0.000 0.000 0.737
Hypercholesterolemia	50 (43.5)		68 (57.1)		0.002
Triglycerides, mg/dL		95 [71–135]		151 [113–190]	0.000
Hypertriglyceridemia	26 (21.8)		62 (52.1)		0.000
Metabolic syndrome	39 (42.4)		53 (57.6)		0.062
eGFR (CKD-EPI), mL/min		72.6 [57–90.3]		68.6 [50.2–96.4]	0.276

Data are expressed as number (percent) or median [interquartile range-IQR].

**Table 4 biomedicines-10-02704-t004:** Metabolic data before and after heart transplant in patients grouped according to changes in glucose metabolism.

	Group A, Worsened (N = 23)		Group B, Improved (N = 31)			Group C, Stably Normal (N = 36)			Group D, Stably Abnormal (N = 29)			
Variables	Pre-tx	Post-tx		Pre-tx	Post-tx		Pre-tx	Post-tx		Pre-tx	Post-tx	
			*p*			*p*			*p*			*p*
Glycemia, mg/dL	112.5 [92–121]	111 [105–124]	0.107	105 [100–123]	91 [85–95]	**0.000**	87 [83.5–92]	88.5 [82.2–94]	0.604	129 [104–157]	108 [95.5–145.5]	0.104
Cholesterol, mg/dL Total *LDL* *HDL*	176 [154–212] 111.5 [90–147.3] 44 [37–54.7]	210 [167.2–256.2] 105.3 [76–124.7] 52.5 [44–65.7]	0.692 0.093 **0.035**	153 [124–187] 97 [71.5–116] 44 [35.5–50.5]	180 [157–214] 103.6 [68.4–125.2] 53 [42.5–67.5]	**0.015** 0.290 **0.004**	144.5 [122.5–167.2] 87 [66–114] 44 [35–56]	173.5 [141.5–224] 97.6 [71.9–122.9] 57 [46.2–67.7]	**0.005** 0.523 **0.002**	147 [120.5–182] 96.1 [56.7–118] 38 [30.5–48]	177 [150–224] 92.6 [77.2–129] 53 [42–74]	0.107 0.881 **0.003**
*Hypercholesterolemia*	9 (42.9)	17 (81)	0.422	17 (54.8)	19 (61.3)	**0.012**	7 (20.6)	13 (38.2)	0.387	17 (58.6)	17 (58.6)	0.471
Triglycerides, mg/dL	103.5 [73.2–168.7]	172 [123–255]	**0.023**	97 [61–151]	152 [104–190]	**0.002**	82.5 [68.2–125]	124.5 [90.2–190.2]	**0.003**	101 [82.5–143.5]	156 [123.5–187]	**0.001**
*Hypertriglyceridemia*	7 (30.4)	12 (52.2)	0.752	8 (25.8)	17 (54.8)	0.698	5 (13.9)	16 (44.4)	0.085	6 (20.7)	17 (58.6)	0.669
BMI, kg/m^2^	26.1 [24.3–30]	26.5 [24.9–30]	**0.043**	24 [22–26.3]	24.1 [22–27]	0.249	23.7 [20.8–26.8]	23.6 [21–26.8]	0.888	27.1 [24.5–30]	27.3 [24.7–30]	0.293
Metabolic syndrome	7 (30.4)	17 (73.9)	**0.007**	13 (41.9)	6 (19.4)	0.097	4 (11.1)	8 (22.2)	0.343	15 (51.7)	22 (75.9)	0.100
eGFR(CKD-EPI), mL/min	80.1 [69.2–94.5]	68.6 [58.2–91.9]	0.128	67.2 [50.2–87.4]	75.9 [52–99.2]	0.624	73.7 [61–90.6]	80.2 [44.2–98.9]	0.857	69 [50.3–87.9]	54.6 [44.4–77.6]	0.078
Arterial hypertension	6 (26.1)	12 (52.2)	0.130	11 (35.5)	16 (51.6)	0.306	7 (19.4)	12 (33.3)	0.285	11 (37.9)	21 (72.4)	**0.017**

Data are expressed as number (percent) or median [interquartile range-IQR].

## Data Availability

Data are available from the corresponding author upon reasonable request.

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
