# Peer review of "Incidence, Risk Factors and Clinical Implications of Glucose Metabolic Changes after Heart Transplant"

_biomedicines, 2022, doi:10.3390/biomedicines10112704_

Round 1

Reviewer 1 Report

The current manuscript by Durante-Magoni et al.described an interesting observation of the association between post - heart transplant diabetes and major outcome of the procedure.  The manuscript is well-written and the focus of the manuscript fits the scope of the journal; however, I do have following concerns and comments:

1) Introduction and discussion should be expanded, and findings in PTDM in other forms of organ transplant should be described and discussed.  

2) Based on Figure 3, improvement in glucose metabolic state post transplant does not "favorably affect mid-term survival".  

3) It would be nice to describe and discuss whether or not there is difference in terms of the use of specific immunotherapies or other treatments between the 4 subgroups.  

4) Please remove the phrases of "numerically declined or improved", which is overinterpretation of the data.  There is a trending decreased eGFR in "stably abnormal" group between pre- vs. post- transplant.  

5) Has any of the patients put on diabetic medications? 

Author Response

Reviewer 1

The current manuscript by Durante-Mangoni et al. described an interesting observation of the association between post – heart transplant diabetes and major outcome of the procedure.  The manuscript is well-written and the focus of the manuscript fits the scope of the journal; however, I do have following concerns and comments:

1) Introduction and discussion should be expanded, and findings in PTDM in other forms of organ transplant should be described and discussed.  

R: We thank the Reviewer for this comment. Accordingly, we expanded introduction comparing PTDM in different types of organ transplant (see Intro, page 2). We also further discussed this point in the relevant paragraphs (see Discussion).

2) Based on Figure 3, improvement in glucose metabolic state post-transplant does not "favorably affect mid-term survival".  

R: We thank the Reviewer for this comment, as this prompts us to better explain our interpretation of data. Group C patients are those who have improvement of glucose metabolism (i.e. from pre-tx diabetes to post-transplant normal glucose metabolism). These patients should be compared to those in group D, who also have glucose metabolic changes before tx but who remain in this condition after tx. By comparing survival curves, it is clearly shown that group B, after glucose metabolism improvement, show a better survival than group D. This is the reason why we state that ‘improvement of glucose metabolic state after transplant … favorably affected mid-term mortality’. However, in order to make this finding clearer to Readers, we also specified that survival ‘ …. was indeed better than in group D where glucose metabolism remained altered’ (last paragraph of Results).

3) It would be nice to describe and discuss whether or not there is difference in terms of the use of specific immunotherapies or other treatments between the 4 subgroups.

R: We thank the Reviewer for this request. We have now added these data as a Supplemetary Table 1, where we provide detailed presentation of immune suppressive drug regimens in use in the 4 subgroups of patients. As shown, there was no significant difference across subgroups. This has been added to the Results, page 8, first few lines.

4) Please remove the phrases of "numerically declined or improved", which is overinterpretation of the data.  There is a trending decreased eGFR in "stably abnormal" group between pre- vs. post- transplant.  

R: As suggested by the Reviewer, we modified this sentence (page 8, last paragraph).

5) Has any of the patients put on diabetic medications? 

R: Yes, all patients had been counselled and were on a hypoglycemic diet. Some were taking oral antidiabetics or insulin, as clinically indicated. These data have been now added in the Results section (page 6, last few lines).

Reviewer 2 Report

I read this paper with interest. This is a clinically important paper. The authors have described their data with good statistical findings.

Few comments.

1. Can authors clarify how long steroids were used in their cohort. Because it said flat dose of 5 mg at 4 weeks. In their program do authors continue steroids as a maintenance regimen and for how long?

2. Authors at the beginning said secondary diabetes due to steroids, but DM can also occur with Tacro/Cyclo (CNI). Authors should describe the effect of CNI on insulin and glucose metabolism in their discussion.

3. Please describe in methods section: When the glycemic studies are done? Is this done at a standard time point after HT or is it done at any time randomly and varies for the cohort? If the glycemic tests are done when the patients are sick or admitted for treatment of rejection or other complication, this underscores the value of their findings as there are many other factors including sickness, high dose steroid, stress, etc... can be confounding issues for their results.

4. Is the maintenance immunosuppression is same for all patient?

5. Can authors use a multi-variable analysis including all pre-HT and post-HT predictors of poor outcomes and decsribe if hyerglycemia is an independent risk factor for poor survival?

Author Response

Reviewer 2

I read this paper with interest. This is a clinically important paper. The authors have described their data with good statistical findings.

Few comments.

  1. Can authors clarify how long steroids were used in their cohort. Because it said flat dose of 5 mg at 4 weeks. In their program do authors continue steroids as a maintenance regimen and for how long?

R: The steroid regimen has been better detailed in the Methods section (page 3, paragraph 2.3).

  1. Authors at the beginning said secondary diabetes due to steroids, but DM can also occur with Tacro/Cyclo (CNI). Authors should describe the effect of CNI on insulin and glucose metabolism in their discussion.

R: We thank for the comment we agree this was needed and have added a description of the role of CNIs and steroids after organ transplant (see Discussion, page 10-11).

  1. Please describe in methods section: When the glycemic studies are done? Is this done at a standard time point after HT or is it done at any time randomly and varies for the cohort? If the glycemic tests are done when the patients are sick or admitted for treatment of rejection or other complication, this underscores the value of their findings as there are many other factors including sickness, high dose steroid, stress, etc... can be confounding issues for their results.

R: We thank the Reviewer for this interesting comment. We specified the follow-up protocol for the glycemic assessment in the Methods (page 3 and in the Results sections (page 4, first 2 lines).

  1. Is the maintenance immunosuppression is same for all patient?

R: The steroids maintenance was the same for all patients, except for specific complications including decompensated diabetes and vertebral fractures. The dosage of immunosuppressive drugs was generally adapted on the bases of Therapeutic Drug Monitoring (TDM) and on white blood cells counts. This has been explained in the Methods section (page 3, para 2.3).

  1. Can authors use a multi-variable analysis including all pre-HT and post-HT predictors of poor outcomes and describe if hyperglycemia is an independent risk factor for poor survival?

R: A multivariable analysis of predictors of mortality was performed and its output is shown in detail in Suppl Table 2. There were no independent predictors of mortality, including hyperglycemia. These data have been added to the results, page 9, last paragraph.

Round 2

Reviewer 2 Report

Thank you for the revision.